# Tetrahydropyridines’ Stereoselective Formation, How Lockdown Assisted in the Identification of the Features of Its Mechanism

**DOI:** 10.3390/molecules27144367

**Published:** 2022-07-07

**Authors:** Anatoly N. Vereshchagin, Taigib M. Iliyasov, Kirill A. Karpenko, Radmir N. Akchurin, Mikhail E. Minyaev

**Affiliations:** N. D. Zelinsky Institute of Organic Chemistry, Russian Academy of Sciences, 47 Leninsky Prospect, 119991 Moscow, Russia; nfsmwm5@mail.ru (T.M.I.); karpenkok_09@mail.ru (K.A.K.); akch.r@yandex.ru (R.N.A.); mminyaev@ioc.ac.ru (M.E.M.)

**Keywords:** multicomponent reactions, domino processes, stereoselectivity, aldehydes, C-H acids, 1,4,5,6-tetrahydropyridines, 3,4,5,6-tetrahydropyridines, 2-aryl-2-hydroxypiperidines, reaction monitoring

## Abstract

The multicomponent reaction of aldehydes, cyano-containing C-H acids, esters of 3-oxocarboxylic acid and ammonium acetate led to unexpected results. The boiling of starting materials in methanol for one to two hours resulted in the formation of polysubstituted 1,4,5,6-tetrahydropyridines with two or three stereogenic centers. During the 2020 lockdown, we obtained key intermediates of this six-step domino reaction. A number of fast and slow reactions occurred during the prolonged stirring of the reaction mass at rt. Sequence: 1. Knoevenagel condensation; 2. Michael addition; 3. Mannich reaction; 4. cyclization—fast reactions and cyclization of the product polysubstituted 2-hydroxypiperidine—was isolated after 40 min stirring at rt. Further monitoring proved the slow dehydration of 2-hydroxypiperidine to obtain 3,4,5,6-tetrahydropyridine after 7 days. Then, four-month isomerization occurred with 1,4,5,6-tetrahydropyridine formation. All reactions were stereoselective. Key intermediates and products structures were verified by X-ray diffraction analysis. Additionally, we specified conditions for the selective intermediates’ preparation.

## 1. Introduction

Six-membered heterocycles form the main subgroup of nitrogen-containing heterocycles. These compounds are well-known frameworks (piperidine, tetrahydropyridine, 1,4-dihydropyridine and pyridine) with a wide spectrum of biological activities [1,2]. Thus, piperidine derivatives display antihypertensive [3], neuroprotective [4,5], antibacterial [6], anticonvulsant [7] and anti-inflammatory [8] abilities, and are inhibitors of farnesyl transferase [9]. Additionally, substituted piperidines are important therapeutic agents in the treatment of influenza [10,11,12], diabetes [13,14], viral infections including AIDS [15,16,17], pulmonary embolism [18] and cancer metastases [19]. Tetrahydropyridines are known as insecticides [20], analgesics [21] and antimalarial agents [22]. Among medications, 4-phenylpiperidine derivatives are of great importance, because they resemble morphine pharmacophore [23,24].

The most common synthetic approaches to produce tetrahydropyridines include imines, in which the nitrogen atom is a source to construct nitrogen-containing six-membered rings. Diels–Alder reactions [25,26], using azadienophiles or azadienes, and domino addition–cyclization reactions involving imines were reported [27,28]. The latter reactions are multicomponent. For organic compounds’ preparation, domino and multicomponent syntheses are superior to two-component reactions in high atom efficiency [29,30], time, materials, energy saving, eco-friendliness and access to greater diversity [31,32,33,34,35,36]. Several publications describe the multicomponent synthesis of substituted tetrahydropyridines from aromatic aldehydes, C-H acids and aromatic amines [37,38], cyanoacetamide [39] or cyanothioacetamide [40].

We have carried out the multicomponent synthesis of substituted piperidines [41,42,43,44,45]. Ammonium acetate or aqueous ammonia were the nitrogen sources for piperidine cycles. Using this approach, we performed the stereoselective synthesis of substituted 1,4,5,6-tetrahydropyridines from electron-deficient olefins and aqueous ammonia [46] (three-component synthesis), or from alkylidenemaloninitriles, 3-oxopropanecarboxylates, aldehydes and ammonium acetate [47] (four-component synthesis) as a nitrogen source for the newly formed six-membered ring. The reaction was carried out by refluxing the starting compounds in methanol for 2–12 h. We hypothesized that the formation of tetrahydropyridine occurs through the following sequence of reactions: Michael addition to obtain 2-substituted 3-aryl-4,4-dicyanobutanoic acid ester **A**, Mannich reaction to give 2-substituted 5-amino-3,5-diaryl-4,4-dicyanopentanoic acid **B**, intramolecular cyclization yielding polysubstituted 2-hydroxypiperidine **C**, and dehydration (Figure 1). Previously, Verboom et al. [48] studied the formation of close analogues of intermediate **A** from benzylidenemalononitriles and malononitrile, or ethyl cyanoacetate. However, the intermediates **B** and **C** have never been isolated or identified.

Subsequently, we studied the multicomponent synthesis of cyclic [49,50,51] and heterocyclic [52,53,54,55,56] compounds from carbonyls and C-H acids. The current research is dedicated to the pseudo-five-component synthesis of 1,4,5,6-tetrahydropyridines **4**,**5** directly from aromatic aldehydes **1** (both with electron-withdrawing and electron-donating substituents), cyano C-H acids **2** (malononitrile or ethyl cyanoacetate), esters of 3-oxocarboxylic acids **3** and ammonium acetate (Figure 2, Table 1). Additionally, we examine the multicomponent process mechanism.

## 2. Results and Discussion

The refluxing of the starting compounds in MeOH led to the selective formation of esters of 2-alkyl(or aryl)-4,6-diaryl-5,5-dicyano-1,4,5,6-tetrahydropyridine-3-carboxylic acids **4** (X=CN) with two stereogenic centers, or diesters of 5-cyano-2,4,6-triaryl-1,4,5,6-tetrahydropyridine-5,3-carboxylates **5** (X=COOEt) with three stereogenic centers (Figure 2, Table 1). This technique was developed in the study of the four-component synthesis of 1,4,5,6-tetrahydropyridines [47]. The new multicomponent reaction allows us to obtain 1,4,5,6-tetrahydropyridines **4, 5** in moderate to excellent yields in one step from cheap and available starting materials via the domino process with the formation of three C-C and two C-N bonds. All reactions were monitored via thin-layer chromatography (TLC). Product **4** was isolated in a 44–90% yield by simple filtration after freezing the reaction mixture. Product **5** was isolated by chromatography in moderate yields of 33% and 36%.

As the NMR spectra of compounds **4**, **5** showed only a single set of signals, we assumed the stereoselective formation of individual diastereoisomers. The structure of **4r** is shown in Figure 1. X-ray crystal diffraction data indicated that structure **4a** with two stereogenic centers should be defined as ethyl (4*SR*,6*RS*)-5,5-dicyano-2-phenyl-4,6-bis(4-methoxy)phenyl-1,4,5,6-tetrahydropyridine-3-carboxylate. The structure of **5a** is shown in Figure 2. X-ray indicated that structure **5a** with three stereogenic centers had the conformation 4*RS*,5*SR*,6*RS*. In both **4r**, **5a** diastereomers, we observed bulky aryl substituents in sterically least-hindered positions relative to each other.

To validate the proposed mechanism (Figure 1), we monitored the reaction between aldehydes **1**, malononitrile **2a**, aryl containing esters of 3-oxocarboxylic acids **3** and ammonium acetate in methanol at room temperature (Figure 3, Table 2). In all cases, 40–45 min stirring of the reaction mass created a dense white precipitate. After filtration and drying, single compounds (by TLC and NMR) were obtained. The ^1^H and ^13^C NMR spectra of compounds **6** showed one set of signals, indicating the formation of a single diastereomer. The **6d** structure is shown in Figure 3. X-ray crystal diffraction data indicated that the structure **6d** with four stereogenic centers should be defined as methyl (2*SR*,3*RS*,4*SR*,6*RS*)-5,5-dicyano-2-(4-bromo)phenyl-2-hydroxy-4,6-bis(4-methyl)phenyl-piperidine-3-carboxylate.

It should be noted that the introduction of alkyl-substituted esters of 3-oxocarboxylic acid **3** (R^1^ = Alk) into this reaction did not result in the formation of 2-hydroxypiperidine **6**. Apparently, the aryl substituent in position 2 is a “stabilizer” of the molecule as a whole. Thus, we found that 2-hydroxypiperidines **6** are formed as a result of a “fast” domino sequence: Knoevenagel condensation, Michael addition, Mannich reaction and intramolecular cyclization. This sequence of reactions takes only 40 min at rt. Unordinary results were found when one of the reaction mixtures was left for a long time without stirring due to isolation measures in spring 2020. The TLS monitoring of the reaction mixture containing 4-methylbenzaldehyde **1d**, malononitrile **2a**, methyl 3-(4-bromophenyl)-3-oxopropanoate **3f** and ammonium acetate in methanol after one and a half months of standing at rt showed the presence of a new substance, different (according to TLS) from 2-hydroxypiperidine **6d** and the final 1,4,5,6-tetrahydropyridine **4t**. We monitored this reaction for 4.5 months. Every week, we took samples of the precipitate from the reaction mixture and analyzed it with ^1^H NMR spectroscopy (Figure 4).

We found the complete conversion of 2-hydroxypiperidine **6d** within a week. A set of signals of the unknown compound **7** and 1,4,5,6-tetrahydropyridine **4t** was observed in the reaction mixture precipitate. Further, over the course of 4 months, we observed the slow transformation of **7** into **4t**. To isolate compound **7**, we made the following assumption. Under reaction conditions, ammonium acetate dissociated into ammonia and acetic acid. Ammonia was consumed to form a six-membered nitrogen-containing ring, while acetic acid remained in the reaction mixture. Therefore, the dehydration of **6d** to **7t** should be carried out under acidification. The acidity of the reaction medium should influence the course of dehydration (Figure 4, Table 3). Indeed, **6d** refluxing in methanol for 2 h in the absence of the acid produced no conversion (Table 3, entry 1). When acidified with 2 eq. of acetic acid, the compound **6d** was completely consumed after 2 h refluxing (Table 3, entry 2, TLC monitoring, eluting with hexane—ethyl acetate, 3:1). Additionally, we observed the formation of **7**. The increase in the acid amount led to **4t** (Table 3, entries 4–7).

The structure of **7** is shown in Figure 5a. X-ray crystal diffraction data indicated that structure **7** with three stereogenic centers should be defined as methyl (3*RS*,4*SR*,6*RS*)-5,5-dicyano-2-(4-bromo)phenyl-4,6-bis(4-methyl)phenyl-3,4,5,6-tetrahydropyridine-3-carboxylate. Structure **7** is the isomer of **4t**. The Gibbs free energy of **4t** is 15.64 kJ/mol lower than that of **7**. DFT calculations were performed with the Gaussian 16 Rev C.01 quantum chemistry program [57].

Likewise, we studied the multicomponent reaction between 4-flurobenzaldehyde **1h**, ethyl cyanoacetate **2b**, methyl 3-(4-bromophenyl)-3-oxopropanoate **3f** and ammonium acetate in methanol at rt (Figure 5). TLC and ^1^H NMR monitoring allowed us to obtain the reaction intermediate **8** in 24%. The structure of **8** is shown in Figure 5b. X-ray crystal diffraction data indicated that structure **8** with four stereogenic centers should be defined as 5-ethyl 3-methyl (3*SR*,4*RS*,5*SR*,6*SR*)-6-(4-bromophenyl)-3-cyano-2,4-bis(4-fluorophenyl)-3,4,5,6-tetrahydropyridine-3,5-dicarboxylate. When ethyl cyanoacetate was introduced into the multicomponent reaction, the formation of an intermediate substituted 2-hydroxypiperidine was not observed.

Crystal data and the structure refinement of **4r**, **5a**, **6d**, **7** and **8** are shown in Table 4.

Thus, the multicomponent reaction between aldehyde **1**, cyano C-H acids **2** (malononitrile or ethyl cyanoacetate), esters of 3-oxocarboxylic acids **3** and ammonium acetate is a six-step domino process (Figure 6). At the first stage, the Knoevenagel condensation between aldehydes and cyano C-H acid occurs. Ammonium acetate is a catalyst for this reaction. The formation of cyano olefins **A** under ammonium salts catalysis is already known [58]. The second step of the process is the Michael addition of C-H acid **3** to the electron-poor styrene **A** to form the Michael adduct **B**. The formation of close analogues of intermediate **B** from benzylidenemalononitriles and malononitrile or ethyl cyanoacetate was studied previously by Verboom et al. [48]. The subsequent Mannich reaction of **B**, aldehyde **1** (second equivalent) and ammonia, which is formed from ammonium acetate, leads to intermediate **C**. The latter undergoes intra-molecular cyclization with the formation of a substituted 2-hydroxypiperidine **6**, which was identified and characterized in this work for the first time. A similar sequence of Knoevenagel condensation—Michael addition—Mannich reaction—intramolecular cyclization was described by Latypova et al. when studying the multicomponent reaction between 1,3-dicarbonyl compounds (two equiv.), formaldehyde and diamines with the formation of substituted bis-1,2,3,4-tetrahydropyridines [59]. None of the intermediates were isolated. Moreover, we tried to isolate **C** in the course of the work, but failed because in the reaction mass, after 10–30 min from the reaction start, there were many compounds (by TLC) that were almost impossible to isolate due to the rapid reaction rate. Polysubstituted 2-hydroxypiperidines **6** were isolated up to 87% even after stirring at rt for 40 min (see Table 2). The fifth step of the domino process is **C** dehydration. We established formation of 3,4,5,6-tetrahydropyridines **7**, **8**. A final isomerization produces 1,4,5,6-tetrahydropyridines **4**, **5**.

## 3. Experimental Procedure

### 3.1. General Information

All melting points were measured with a Stuart SMP30 melting point apparatus (Bibby Sterling Ltd., Granton, UK). ^1^H and ^13^C NMR spectra were recorded with a Bruker AM300 (Bruker, Bremen, Germany) and Bruker DRX 500 (Bruker BioSpin GmbH, Bremen, Germany) at ambient temperature in DMSO-d_6_ or CDCl_3_ solutions. Chemical shifts values are given in δ scale relative to Me_4_Si. The *J* values are given in hertz. Only discrete or characteristic signals for the 1H NMR are reported. IR spectra were recorded with a Bruker ALPHA-T FT-IR spectrometer (Bruker Corporation, Bremen, Germany) in KBr pellets. HR-ESI-MS were measured on a Bruker microTOF II instrument (Bruker Daltonik GmbH, Bremen, Germany); external or internal calibration was performed with electrospray calibrant solution (Fluka). All starting materials were obtained from commercial sources and used without purification. All reactions were monitored with thin-layer chromatography (TLC) and carried out with Merck precoated plates DC-AlufolienKieselgel60 F254 (Merck KGaA, Darmstadt, Germany). X-ray crystallographic analyses were performed with Bruker Quest D8 diffractometer (Bruker AXS GmbH, Bremen, Germany). 

### 3.2. DFT Calculations

DFT calculations were performed with Gaussian 16 Rev C.01. B3LYP DFT (Gaussian Inc., Wallingford CT, USA, 2016) functional with GD3BJ empirical dispersion correction, and a Def2SVP basis set was used for geometry optimization and calculations of thermodynamics. Data from X-ray diffraction experiment for **7** were used as starting points for geometry optimizations. Cartesian coordinates are given in angstroms; absolute energies for all substances are given in hartrees. The analysis of vibrational frequencies was performed for all optimized structures. All compounds were characterized by only real vibrational frequencies. Wavefunction stability, using stable keyword, was also checked for each molecule. For more information see Appendix A.

For the calculations of the optimized geometries, frequencies and thermodynamics with the following keywords were used:*# opt freq b3lyp nosymm def2svp empiricaldispersion = gd3bj test*

### 3.3. X-ray Crystallographic Data and Refinement Details

X-ray diffraction data for all compounds were collected at 100 K on a Bruker Quest D8 diffractometer equipped with a Photon-III area detector, using graphite-monochromatized Mo Kα-radiation (0.71073 Å) and the shutterless φ- and ω-scan technique. Relying on the analysis of preliminary collected reflections with the Cell_Now program [60], all crystals of **8** from various batches contained over seven major domains with apparently chaotic orientations. This, along with a chiral space group, seriously impeded data analysis, resulting in six attempts to collect reflection data and to solve the structure. The intensity data were integrated by the SAINT program [61] and were semi-empirically corrected for absorption and decay, using SADABS [62] for **4r**, **6d**, **5a** and **7** or using TWINABS [61] for **8**. The structures were solved by direct methods using SHELXT [63] and refined by the full-matrix least-squares method on *F^2^* using SHELXL-2018 [64]. The crystals of **6d** and **7** were refined as inversion twins, for which the absolute structure parameter (Flack) was determined by classical fit [65]. The selected specimen of **8** was refined as a non-merohedral 2-component twin.

All non-hydrogen atoms were refined with individual anisotropic displacement parameters. The locations of atoms H1 (in **4r**, **5a**) and H1A, H1B (in **6d**) were found from the electron density difference map; these H atoms were refined with individual isotropic displacement parameters. All other hydrogen atoms were placed in geometrically calculated positions and refined as riding atoms with relative isotropic displacement parameters. A rotating group model was applied for methyl groups. Mercury program [66] was used for molecular graphics. Crystal data, data collection and structure refinement details are summarized in Table 4.

### 3.4. Synthesis of ***4***–***5***

A mixture of aldehydes **1** (6 mmol), cyano C-H acid **2** (3 mmol), ester of 3-oxocarboxylic acids **3** (3 mmol) and ammonium acetate (6 mmol) was refluxed in methanol (10 mL) for 2 h. After the reaction completion, the mixture was maintained at –10 °C for 30 min for the complete precipitation of the product, the precipitate was collected by filtration and dried to obtain pure tetrahydropyridine **4**. Compound **5** was purified by column chromatography.

Methyl (4SR,6RS)-5,5-dicyano-2-methyl-4,6-diphenyl-1,4,5,6-tetrahydropyridine-3-carboxylate (4a) Yield: 0.86 g (80%); white solid; m.p. 218–219 °C. (lit. [46] m.p. 218–219 °C); 1H-NMR (DMSO-*d*_6_, 300.13 MHz): δ = 7.63–7.7 (m, 2H, Ar), 7.52 (dd, 4 H, Ar, *J*_1_ = 5.9 Hz, *J*_2_ = 1.6 Hz), 7.34–7.28 (m, 4H, Ar + NH), 5.27 (s, 1H, CH), 4.83 (s, 1H, CH), 3.11 (s, 3H, OCH3), 2.32 (s, 3H, CH3).

Methyl (4SR,6RS)-5,5-dicyano-2-methyl-4,6-bis(2-methyl)phenyl-1,4,5,6-tetrahydropyridine-3-carboxylate (**4b**) Yield: 0.75 g (65%); white solid; m.p. 233–235 °C; 1H-NMR (CDCl3, 300.13 MHz): δ = 7.92–7.86 (m, 1H, Ar), 7.51–7.18 (m, 7H, Ar), 5.22 (s, 1H, CH), 5.09 (s, 1H, CH), 4.35 (s, 1H, NH), 3.26 (s, 3H, OCH3), 2.59 (s, 3H, CH3), 2.56 (s, 3H, CH3), 2.4 (s, 3H, CH3); 13C-NMR (CDCl3, 75.47 MHz): δ = 166.67, 151.99, 137.24, 136.92, 135.95, 131.95, 131.47, 130.69, 130.20, 128.03, 127.29, 126.49, 126.34, 126.22, 113.82, 112.61, 99.28, 57.54, 50.44, 46.09, 45.63, 20.04, 19.89; IR (KBr): 3343, 2249, 1686, 1460, 1247 cm^−1^; HRMS (ESI) *m*/*z* [M + H]+ calcd for C24H24N3O2^+^: 386.1863; found: 386.1857.

Methyl (4SR,6RS)-5,5-dicyano-2-methyl-4,6-bis(3-methyl)phenyl-1,4,5,6-tetrahydropyridine-3-carboxylate (**4c**) Yield: 0.88 g (76%); white solid; m.p. 191–193 °C; 1H-NMR (CDCl3, 300.13 MHz): δ = 7.11–7.48 (m, 8H, Ar), 4.72 (s, 1H, CH), 4.57 (s, 1H, CH), 4.43 (s, 1H, NH), 3.30 (s, 3H, OCH3), 2.44 (s, 3H, CH3), 2.42 (s, 3H, CH3), 2.38 (s, 3H, CH3); 13C-NMR (CDCl3, 75.47 MHz): δ = 166.92, 151.88, 139.25, 138.11, 137.65, 133.55, 131.46, 129.23, 129.17, 128.47, 128.39, 124.96 (s, 2C), 113.77, 111.91, 97.84, 61.92, 51.05, 50, 51, 47.99, 21.48 (s, 2C), 20.29; IR (KBr): 3422, 2252, 1655, 1453, 1248 cm−1; HRMS (ESI) *m*/*z* [M + H]+ calcd for C24H24N3O2^+^: 386.1863; found: 386.1857.

Methyl (4SR,6RS)-5,5-dicyano-2-methyl-4,6-bis(4-methyl)phenyl-1,4,5,6-tetrahydropyridine-3-carboxylate (**4d**) Yield: 0.83 g (72%); white solid; m.p. 208–210 °C. (lit. [47] m.p. 208–210 °C); 1H-NMR (DMSO-*d*_6_, 300.13 MHz): δ = 7.51 (d, *J* = 8.0 Hz, 2H, Ar), 7.44 (s, 1H, NH), 7.32 (d, *J* = 8.0 Hz, 2H, Ar), 7.21–7.14 (m, 4H, Ar), 5.19 (s, 1H, CH), 4.75 (s, 1H, CH), 3.13 (s, 3H, OCH3), 2.36 (s, 3H, CH3), 2.30 (s, 6H, 2CH3).

Methyl (4SR,6RS)-5,5-dicyano-2-methyl-4,6-bis(3-fluoro)phenyl-1,4,5,6-tetrahydropyridine-3-carboxylate (**4e**) Yield: 0.84 g (71%); white solid; m.p. 174–176 °C. (lit. [47] m.p. 174–176 °C); 1H-NMR (DMSO-d6, 300.13 MHz): δ = 7.7–7.13 (m, 8H, Ar + NH), 7.06 (d, *J* = 10.11 Hz, 1H, Ar), 5.32 (s, 1H, CH), 4.87 (s, 1H, CH), 3.17 (s, 3H, CH3), 2.34 (s, 3H, CH3).

Methyl (4SR,6RS)-5,5-dicyano-2-methyl-4,6-bis(3-chloro)phenyl-1,4,5,6-tetrahydropyridine-3-carboxylate (**4f**) Yield: 0.87 g (68%); white solid; m.p. 210–213 °C; 1H-NMR (DMSO-*d*_6_, 300.13 MHz): δ = 7.68 (d, *J* = 10.15 Hz, 2H, Ar), 7.63–7.25 (m, 7H, Ar + NH), 5.31 (s, H, CH), 4.86 (s, H, CH), 3.18 (s, 3H, OCH3), 2.34 (s, 3H, CH3); 13C-NMR (CDCl3, 75.47 MHz): δ = 166.39, 152.75, 139.74, 135.36, 135.19, 134.46, 131.06, 130.68, 129.9, 128.8, 128.07 (s, 2C), 126.27, 125,86, 113.22, 111.36, 97, 19, 61.22, 50.68, 50.46, 47, 48, 20.35; IR(KBr): 3411, 2240, 1712, 1458, 1247, 711 cm^−1^; HRMS (ESI) *m*/*z* [M + H]+ (for 35Cl) 426.0760 calcd for C22H18Cl2N3O2^+^: 426.0771, (for 35Cl and 37Cl) 428.0736 calcd for C22H18Cl2N3O2^+^: 428.0742.

Methyl (4SR,6RS)-5,5-dicyano-2-methyl-4,6-(3-pyridine)-1,4,5,6-tetrahydropyridine-3-carboxylate (**4g**) Yield: 0.67 g (62%); white solid; m.p. 200–203 °C; 1H-NMR (DMSO-*d*_6_, 300.13 MHz): δ = 8.81 (d, *J* = 8 Hz, 1H, Ar), 8.72 (dd, *J* = 4.8 Hz, *J*_1_ = 1.5 Hz, 1H, Ar), 8.56–8.53 (m, 2H, Ar), 8.03 (dt, *J* = 8 Hz, *J*_1_ = 1.7 Hz, 1H, Ar), 7.8 (s, 1H, NH), 7.68 (dt, *J* = 8 Hz, *J*_1_ = 1.6 Hz, 1H, Ar), 7.6 (dd, *J* = 8 Hz, *J*_1_ = 4.8 Hz, 1H, Ar), 7.43 (dd, *J* = 8 Hz, *J*_1_ = 4.8 Hz, 1H, Ar), 5.42 (s, 1H, CH), 4.95 (s, 1H, CH), 3.16 (s, 3H, OCH3), 2.36 (s, 3H, CH3); 13C-NMR (DMSO-*d*_6_, 125.76 MHz): δ = 166.01, 153.02, 152.28, 149.94, 149.60, 149.48, 135.23 (s, 2C), 133.51, 129.22, 124.2, 123.55, 112.85, 113.30, 96.97, 59.70, 50.82, 48.51, 47.55, 20.58; IR (KBr): 3204, 2248, 1651, 1459, 1263 cm^−1^; HRMS (ESI) *m*/*z* [M + H]+ calcd for C20H18N5O2^+^: 360.1455; found: 360.1456.

Ethyl (4SR,6RS)-5,5-dicyano-2-methyl-4,6-diphenyl-1,4,5,6-tetrahydropyridine-3-carboxylate (**4h**) Yield: 1.00Γ (90%); white solid; m.p. 200–202 °C; 1H-NMR (DMSO-*d*_6_, 500.13 MHz): δ = 7.72–7.31 (m, 11H, Ar + NH), 5.31 (s, 1H, CH), 4.86 (s 1HCH), 3.73–3.55 (m, 2H CH2), 2.36 (s, 3H, CH3), 0.57 (t, *J* = 7.12Hz, 3H, CH3); 13C-NMR (DMSO-*d*_6_, 75.47 MHz): δ =166.25, 154.33, 139.71, 134.84, 130.43, 129.06 (s, 2C), 128.82 (s, 2C), 128.63 (s, 2C), 128.33 (s, 3C), 114.44, 113.36, 94.9, 60.01, 58.61, 49.64, 48.2, 19.43, 13.79; IR (KBr): 3312, 2252, 1644, 1470, 1456, 1247 cm^−1^; HRMS (ESI) *m*/*z* [M + H]+ calcd for C23H22N3O2^+^: 372.1707; found: 372.1700.

Ethyl (4SR,6RS)-5,5-dicyano-2-methyl-4,6-bis(4-fluoro)phenyl-1,4,5,6-tetrahydropyridine-3-carboxylate (**4i**) Yield: 0.84 g (69%); white solid; m.p. 154–156 °C; 1H-NMR (DMSO-d6, 300.13 MHz): δ = 7.72–7.61 (m, 2H, Ar), 7.55 (s, 1H, NH), 7.44–7.3 (m, 4H, Ar), 7.29–7.16 (m, 2H, Ar), 5.31 (s, 1H, CH), 4.85 (s, 1H, CH), 3.77–3.54 (m, 2H, CH2), 2.32 (s, 3H, CH3), 0.61 (t, *J* = 7.19 Hz, 3H, CH3); 13C-NMR (DMSO-*d*_6_, 75.47 MHz): δ = 166.09, 163.44 (s, *J*^1^_C-F_ = 246.7 Hz, 1C), 162.24 (s, *J*^1^_C-F_ = 244.1Hz, 1C), 154.51, 135.08 (d, *J*^4^_C-F_ = 2.9Hz, 2C), 131.01 (d, *J*^3^_C-F_ = 8.7Hz, 2C), 130.05 (d, *J*^3^_C-F_ = 8.6Hz, 2C), 116.06 (d, *J*^2^_C-F_ = 21.8Hz, 2C), 115.54 (d, *J*^2^_C-F_ = 21.6Hz, 2C), 114.28, 113.23, 94,81, 59.16, 58.69, 48.74, 48.31, 19.47, 13.87; IR (KBr): 3352, 2253, 1688, 1458, 1250, 1158 cm^−1^; HRMS (ESI) *m*/*z* [M + H]+ calcd for C23H20F2N3O2^+^: 408.1518; found: 408.1512.

Ethyl (4SR, 6RS)-5,5-dicyano-2-methyl-4,6-bis(4-nitro)phenyl-1,4,5,6-tetrahydropyridine-3-carboxylate (**4j**) Yield: 0.87Γ (63%); white solid; m.p. 242–243 °C; 1H-NMR (DMSO-*d*_6_, 300.13 MHz): δ = 8.43 (d, *J* = 8.66 Hz, 2H, Ar), 8.30 (d, *J* = 8.69 Hz, 2H, Ar), 7.92 (s, 1H, NH), 7.88 (d, *J* = 5.15 Hz, 2H, Ar), 7.61 (d, *J* = 8.31 Hz, 2H, Ar), 5.54 (s, H, CH), 5.09 (s, H, CH), 3.77–3.58 (m, 2H, OCH2), 0.62 (t, *J* = 7.07 Hz, 3H, CH3); 13C-NMR (DMSO-*d*_6_, 125.76 MHz): δ = 165.68, 155.48, 149.13, 147.75, 147.29, 141.16, 130.40 (s, 2C), 124.29 (s, 4C), 124.07 (s, 2C), 113.62, 112.65, 94.04, 59.04, 58.95, 48.76, 46.99, 19.76, 13.94; IR (KBr): 3372, 2250, 1689, 1348, 1242 cm^−1^; HRMS (ESI) *m*/*z* [M + H]+ calcd for C23H20N5O6^+^: 462.1408; found: 462.1402.

Methyl (4SR,6RS)-5,5-dicyano-2-ethyl-4,6-diphenyl-1,4,5,6-tetrahydropyridine-3-carboxylate (**4k**) Yield: 1.00 g (90%); white solid; m.p. 115–117 °C. (lit. [47] m.p. 115–117 °C); 1H-NMR (CDCl3 400.16 MHz): δ = 7.69–7.31 (m, 10H, Ar), 4.78 (s, 1H, CH), 4.63 (s, 1H, CH), 4.54 (s, 1H, NH), 3.27 (s, 3H, OCH3), 2.94–2.67 (m, 2H, CH2),1.35 (t, *J* = 7.5 Hz, 3H, CH3).

Methyl (4SR,6RS)-5,5-dicyano-2-ethyl-4,6-bis(4-methyl)phenyl-1,4,5,6-tetrahydropyridine-3-carboxylate (**4l**) Yield: 0.98 g (82%); white solid; m.p. 112–115 °C; 1H-NMR (CDCl3 300.13 MHz): δ = 7.53 (d, *J* = 8.1 Hz, 2H, Ar), 7.31 (d, *J* = 7.8 Hz, 4H, Ar), 7.17 (d, *J* = 8 Hz, 2H, Ar), 4.72 (s, 1H, CH), 4.58 (s, 1H, CH), 4.46 (s, 1H, NH), 3.30 (s, 3H, OCH3), 2.92–2.64 (m, 2H, CH2), 2.42 (s, 3H, CH3), 2.35 (s, 3H, CH3), 1.33 (t, *J* = 7.5 Hz, 3H, CH3); 13C-NMR (DMSO-*d*_6_, 75.47 MHz): δ = 166.59, 157.37, 140.84, 138, 134.79, 130.75, 130 (s, 2C), 129.27 (s, 2C), 127.71 (s, 4C), 113.92, 111.94, 61.47, 50.64, 50.53, 48.34, 26.92, 21.28, 21.18, 13.38; IR (KBr): 3354, 2254, 1695, 1466, 1261 cm^−1^; HRMS (ESI) *m*/*z* [M + H]+ calcd for C25H26N3O2^+^: 400.2020; found: 400.2014.

Methyl (4SR,6RS)-5,5-dicyano-2-ethyl-4,6-bis(4-bromo)phenyl-1,4,5,6-tetrahydropyridine-3-carboxylate (**4m**) Yield: 1.30 g (82%); white solid; m.p. 214–217 °C. (lit. [47] m.p. 214–217 °C); 1H-NMR (CDCl3 300.13 MHz): δ = 7.66 (d, *J* = 8.4 Hz, 2H, Ar), 7.54–7.49 (m, 4H, Ar), 7.3 (d, *J* = 8.4 Hz, 2H, Ar), 4.74 (s, 1H, CH), 4.59 (s, 1H, CH), 4.49 (s, 1H, NH), 3.33 (s, 3H, OCH3), 2.93–2.65 (m, 2H, CH2), 1.32 (t, *J* = 7.5 Hz, 3H, CH3).

Methyl (4SR, 6RS)-5,5-dicyano-2-ethyl-4,6-bis(4-nitro)phenyl-1,4,5,6-tetrahydropyridine-3-carboxylate (**4n**) Yield: 0.80 g (58%); white solid; m.p. 243–248 °C; IR (KBr): 3387, 2250, 1685, 1484, 1349, 1256 cm^−1^; 1H-NMR (DMSO-d6, 300.13 MHz): δ = 8.44 (d, *J* = 8.7 Hz, 2H, Ar), 8.3 (d, *J* = 8.8 Hz, 2H, Ar), 7.92 (s, 1H, NH), 7.9 (d, *J* = 8.9 Hz, 2H, Ar), 7.58 (d, *J* = 8.5 Hz, 2H, Ar), 5.52 (s, 1H, CH), 5.08 (s, 1H, CH), 3.16 (s, 3H, OCH3), 2.88–2.65 (m, 2H, CH2), 1.27 (t, *J* = 7.3Hz, 3H, CH3); 13C-NMR (DMSO-*d*_6_, 75.47 MHz): δ = 165.99, 161.28, 149.06, 147.65, 147.19, 141.04, 130.31 (s, 2C), 129.24 (s, 2C), 124.13 (s, 2C), 123.99 (s, 2C), 113.51, 112.45, 92.83, 58.88, 50.51, 48.47, 46.99, 26.19, 14.49; IR (KBr): 3387, 2250, 1685, 1484, 1349, 1256 cm^−1^; HRMS (ESI) *m*/*z* [M + H]+ calcd for C23H20N5O6^+^: 462.1408; found: 462.1401.

Methyl (4SR,6RS)-5,5-dicyano-2–4,6-triphenyl-1,4,5,6-tetrahydropyridine-3-carboxylate (**4o**) Yield: 0.95 g (76%); white solid; m.p. 223–225 °C. (lit. [47] m.p. 223–225 °C); 1H-NMR (CDCl3 300.13 MHz): δ = 7.73–7.33 (m, 15H, Ar), 4.98 (s, 1H, CH),4.78 (s, 1H, CH), 4.61 (s, 1H, NH), 3.15 (s, 3H, OCH3).

Methyl (4SR,6RS)-5,5-dicyano-2-phenyl-4,6-bis(3-fluoro)phenyl-1,4,5,6-tetrahydropyridine-3-carboxylate (**4p**) Yield: 0.69 g (52%); white solid; m.p. 191–193 °C; 1H-NMR (CDCl3 300.13 MHz): δ = 7.55–7.19 (m, 12H, Ar), 7.1 (t, *J* = 8Hz, 1H, Ar), 4.96 (s, 1H, CH), 4.77 (s, 1H, CH), 4.62 (s, 1H, NH), 3.19 (s, 3H, OCH3); 13C-NMR (CDCl3, 125.76 MHz): δ = 165.68, 162.93 (d, *J*^1^_C-F_ = 249.7Hz, 1C), 162.77 (d, *J*^1^_C-F_ = 247Hz, 1C), 154.36, 139.49 (d, *J*^4^_C-F_ = 7.1 Hz, 1C), 136.07, 135.28 (d, *J*^4^_C-F_ = 7.2 Hz, 1C), 131.27 (d, *J*^3^_C-F_ = 8.3 Hz, 1C), 130.34 (d, *J*^3^_C-F_ = 8.3 Hz, 1C), 129.99, 128.38 (s, 2C), 128.34 (s, 2C), 123.8 (d, *J*^6^_C-F_ = 3 Hz, 2C), 118.08 (d, *J*^2^_C-F_ = 21 Hz, 1C), 115.88 (d, *J*^2^_C-F_ = 21.1 Hz, 1C), 115.25, 115.07, 113.21, 111.40, 98.68, 61.60, 50.80, 50.75, 47.64; IR(KBr): 3376, 2250, 1704, 1263, 1105 cm^−1^; HRMS (ESI) *m*/*z* [M + H]+ calcd for C27H20F2N3O2^+^: 456.1518; found: 456.1507.

Ethyl (4SR,6RS)-5,5-dicyano-2-phenyl-4,6-diphenyl-1,4,5,6-tetrahydropyridine-3-carboxylate (**4q**) Yield: 0.75 g (58%); white solid; m.p. 118–121 °C; IR(KBr): 3385, 2240, 1699, 1466, 1260 cm^−1^; 1H-NMR (CDCl3 300.13 MHz): δ = 7.72–7.34 (m, 15H, Ar), 4.98 (s, 1H, CH), 4.79 (s, 1H, CH), 4.6 (s, 1H, NH), 3.75–3.56 (m, 2H, CH2), 0.6 (t, *J* = 7.1 Hz, 3H, CH3); 13C-NMR (CDCl3, 75.47 MHz): δ = 165.37, 153.91, 137.29, 136.54, 133.33, 130.77, 129.66, 129.42 (s, 2C), 128.68 (s, 2C), 128.63, 128.51 (s, 2C), 128.24 (s, 4C), 127.93 (s, 2C), 113.66, 111.85, 99.18, 62.23, 59.53, 51.31, 48.17, 13.3; HRMS (ESI) *m*/*z* [M + H]+ calcd for C28H24N3O2^+^: 434.1863; found: 434.1850.

Ethyl (4SR,6RS)-5,5-dicyano-2-phenyl-4,6-bis(4-methoxy)phenyl-1,4,5,6-tetrahydropyridine-3-carboxylate (**4r**) Yield: 0.65 g (44%); white solid; m.p. 177–179 °C; 1H-NMR (CDCl3 300.13 MHz): δ = 7.58 (d, *J* = 8.6 Hz, 2H, Ar), 7.53–7.42 (m, 7H, Ar), 7 (d, *J* = 8.6 Hz, 2H, Ar), 6.94 (d, *J* = 8.6 Hz, 2H, Ar), 4.92 (s, 1H, CH), 4.72 (s, 1H, CH), 4.52 (s, 1H, NH), 3.84 (s, 3H, OCH3), 3.83 (s, 3H, OCH3), 3.72–3.59 (m, 2H, CH2), 0.64 (t, *J* = 7.1, 3H, CH3); 13C-NMR (CDCl3, 75.47 MHz): δ = 165.53, 161.29, 159.74, 153.50, 136.69, 129.54, 129.28 (s, 2C),129.18, 129.15 (s, 2C), 128.44 (s, 2C), 128.19 (s, 2C), 125.35, 114.74 (s, 2C), 114.06 (s, 2C), 113.94, 112.12, 99.42, 61.70, 59.50, 55.38, 55.22, 50.66, 48.77, 13.39; IR (KBr): 3345, 2256, 1699, 1445, 1252 cm^−1^; HRMS (ESI) *m*/*z* [M + H]+ calcd for C30H28N3O4^+^: 494.2074; found: 494.2062.

Methyl (4SR,6RS)-5,5-dicyano-2-(4-bromo)phenyl-4,6-diphenyl-1,4,5,6-tetrahydropyridine-3-carboxylate (**4s**) Yield: 0.92 g (62%); white solid; m.p. 167–170 °C. (lit. [47] m.p. 167–170 °C); 1H-NMR (CDCl3 300.13 MHz): δ = 7.69–7.33 (m, 14H, Ar), 4.96 (s, 1H, CH), 4.75 (s, 1H, CH), 4.57 (s, 1H, NH), 3.17 (s, 3H, OCH3).

Methyl (4SR,6RS)-5,5-dicyano-2-(4-bromo)phenyl-4,6-bis(4-methyl)phenyl-1,4,5,6-tetrahydropyridine-3-carboxylate (**4t**) Yield: 1.18 g (75%); white solid; m.p. 131–134 °C; 1H-NMR (CDCl3 300.13 MHz): δ = 7.59 (d, *J* = 8.2 Hz, 2H, Ar), 7.53 (d, *J* = 7.9 Hz, 2H, Ar), 7.42 (d, *J* = 6.3 Hz, 2H, Ar), 7,39 (d, *J* = 6.6 Hz, 2H, Ar), 7.3 (d, *J* = 9 Hz, 2H, Ar), 7.22 (d, *J* = 7.8 Hz, 2H, Ar), 4.90 (s, 1H, CH), 4.70 (s, 1H, CH), 4.54 (s, 1H, NH), 3.18 (s, 3H, OCH3), 2.41 (s, 3H, CH3), 2.38 (s, 3H, CH3); 13C-NMR (DMSO-*d*_6_, 75.47 MHz): δ =165.99, 154.72, 135.77, 136.24, 137.72, 139.86, 131.48 (s, 2C), 130.98 (s, 2C), 129.4 (s, 2C), 129.34 (s, 2C), 128.83 (c, 2C),128.13 (c, 3C), 122.64, 114.51, 113.34, 97.06, 60.43, 50.46, 49.47, 48.82, 21.26, 21.2; IR (KBr): 3484, 2255, 1690, 1433, 1262, 726 cm^−1^; HRMS (ESI) *m*/*z* [M + H]+ (for 79Br) 526.1128 calcd for C29H25BrN3O2^+^: 526.1125.

5-ethyl 3-methyl (4RS,5SR,6RS)-5-cyano-2-(4-bromo)phenyl-4,6-bis(4-bromo)phenyl-1,4,5,6-tetrahydropyridine-5,3-carboxylate (**5a**) Yield: 0.77 g (36%); white solid; m.p. 221–223 °C; 1H-NMR (CDCl3 300.13 MHz): δ = 7.62–7.18 (m, 12H, Ar), 4.87 (s, 1H, CH), 4.82 (s, 1H, CH),4.5 (s, 1H, NH), 3.93 (q, *J* = 7 Hz, 2H, OCH2), 3.17 (s, 3H, OCH3), 0.89 (t, *J* = 7,1 Hz, 3H, CH3); 13C-NMR (CDCl3, 75.47 MHz): δ = 166.14, 166.09, 152.55, 137.52, 135.69, 133.13, 132.33 (s, 2C),131.71, 131.59 (s, 2C), 131.44 (s, 2C), 130.14 (s, 2C), 129.46 (s, 2C), 129.34, 124.35, 123.86, 121.8, 114.87, 100.12, 63.14, 61.31, 57.65, 50.63, 49.16, 13.6; IR(KBr): 3334, 2247, 1737, 1259, 501 cm^−1^; HRMS (ESI) *m*/*z* [M + H]+ (for 79Br) 700.9281 calcd for C29H24Br3N2O4^+^: 700.9276.

5-ethyl 3-methyl (4RS,5SR,6RS)-5-cyano-2-(4-chloro)phenyl-4,6-bis(4-bromo)phenyl-1,4,5,6-tetrahydropyridine-5,3-carboxylate (**5b**) Yield: 0.61 g (33%); white solid; m.p. 205–208 °C; IR (KBr): 3333, 2247, 1739, 1259, 810, 500 cm^−1^; 1H-NMR (CDCl3 300.13MHz): δ = 7.59–7.21 (m, 12H, Ar), 4.87 (s, 1H, CH), 4.83 (s, 1H, CH), 4.48 (s, 1H, NH), 3.93 (q, *J* = 7 Hz, 2H, OCH2), 3.17 (s, 3H, OCH3), 0.89 (t, *J* = 7,1 Hz, 3H, CH3); 13C-NMR (CDCl3, 75.47 MHz): δ = 166.17, 166.1, 152.56, 137.55, 135.65, 135.18, 133.14, 132.34 (s, 2C), 131.6 (s, 2C), 129.91 (s, 2C), 129.47 (s, 2C), 129.33 (s, 2C), 128.48 (s, 2C), 124.35, 121.8, 114.89, 100.1, 63.15, 61.30, 57.67, 50.63, 49.15, 13.61; IR (KBr): 3333, 2247, 1739, 1259, 810, 500 cm^−1^; HRMS (ESI) *m*/*z* [M + H]+ (for 35Cl and 79Br) 656.9786 calcd for C29H24Br2ClN2O4^+^: 656.9776.

### 3.5. Synthesis of ***6***

A mixture of aldehyde **1** (6 mmol), malononitril **2a** (3 mmol), ester of 3-oxocarboxylic acid **3** (3 mmol) and ammonium acetate (6 mmol) was stirred in methanol (7 mL) at rt for 40 min. The precipitate was collected by filtration and dried to obtain piperidine **6**.

Ethyl (2SR,3RS,4SR,6RS)-5,5-dicyano-2-phenyl-2-hydroxy-4,6-diphenyl-piperidine-3-carboxylate (**6a**) Yield: 0.97 g (72%); white solid; m.p. 135–137 °C; 1H-NMR (DMSO-*d*_6_, 300.13 MHz): 7.72 (d, *J* = 7.2 Hz, 4H, Ar), 7.5–7.3 (m, 11H, Ar), 6.01 (s, OH), 5.14 (s, H, CH), 4.39 (d, *J* = 12.4 Hz, H, CH), 3.58 (s, NH), 3.52 (q, *J* = 7.1 Hz, 2H, CH2), 3.45 (d, *J* = 12.4 Hz, H, CH), 0.56 (t, *J* = 7.1 Hz, 3H, CH3); 13C-NMR (DMSO-*d*_6_, 75.47 MHz): 168.2, 144.7, 136.9, 136.2, 130.06, 129.3, 129.04 (s, 4C), 128.9 (s, 4C), 128.4, 128.3 (s, 2C), 126.5 (s, 2C), 114.1, 113.5, 84.4, 60.12, 59.5, 54.7, 49.2, 46.7, 13.7; IR (υmax) (KBr), ν/cm^−1^: 3503, 3317, 1711, 703 cm^−1^; HRMS (ESI) *m*/*z* [M + H]+ 452.1969 calcd for C28H26N3O3^+^: 452.1977.

Ethyl (2SR,3RS,4SR,6RS)-5,5-dicyano-2-phenyl-2-hydroxy-4,6-bis(4-methyl)phenyl-piperidine-3-carboxylate (**6b**) Yield: 0.88 g (61%); white solid; m.p. 130–132 °C; 1H-NMR (DMSO-*d*_6_, 300.13 MHz): 7.71 (d, *J* = 7.1 Hz, 2H, Ar), 7.59 (d, *J* = 8.1 Hz, 2H, Ar), 7.45–7.24 (m, 7H, Ar), 7.19 (d, *J* = 8.1 Hz, 2H, Ar), 6 (s, OH), 5.06 (s, H, CH), 4.31 (d, *J* = 12.4 Hz, H, CH), 3.51 (q, *J* = 7.1 Hz, 2H, CH2), 3.47 (s, NH), 3.41 (d, *J* = 12.4 Hz, H, CH), 2.31 (s, 3H, CH3), 2.28 (s, 3H, CH3), 0.58 (t, *J* = 7.1 Hz, 3H, CH3); 13C-NMR (DMSO-d6, 75.47MHz): 168.2, 144.8, 139.5, 138.6, 134, 133.3, 129.6 (s, 2C), 129.4 (s, 2C), 128.9 (s, 2C), 128.7 (s, 2C), 128.4, 128.3 (s, 2C), 126.5 (s, 2C), 114.2, 113.5, 84.4, 60.1, 59.2, 54.7, 49.5, 46.3, 21.3, 21.1, 13.7; IR (υmax) (KBr), ν/cm^−1^: 3498, 3320, 2224, 1713, 702 cm^−1^; HRMS (ESI) *m*/*z* [M + H]+ 480.2293 calcd for C30H30N3O3^+^: 480.2282.

Ethyl (2SR,3RS,4SR,6RS)-5,5-dicyano-2-phenyl-2-hydroxy-4,6-bis(4-chloro)phenyl-piperidine-3-carboxylate (**6c**) Yield: 0.87 g (56%); white solid; m.p. 126–128 °C; 1H-NMR (DMSO-*d*_6_, 300.13 MHz): δ = 7.72 (d, *J* = 8.6 Hz, 2H, Ar), 7.73–7.69 (m, 2H, Ar), 7.56 (d, *J* = 8.5 Hz, 2H, Ar), 7.48 (d, *J* = 2.4 Hz, 3H, Ar), 7.52–7.31 (m, 4H, Ar), 6.07 (s, OH), 5.17 (s, H, CH), 4.46 (d, *J* = 12.4 Hz, H, CH), 3.75 (s, NH), 3.54 (q, *J* = 7.1 Hz, 2H, CH2), 3.42 (d, *J* = 12.3 Hz, H, CH), 0.59 (t, *J* = 7.1 Hz, 3H, CH3); 13C-NMR (DMSO-*d*_6_, 75.47MHz): δ = 168, 144.5, 135.75, 135.27, 134.7, 134.1, 131.95 (s, 2C), 130.7 (s, 2C), 129.2 (s, 2C), 128.96 (s, 2C), 128.4, 128.3 (s, 2C), 126.6 (s, 2C), 113.8, 113.2, 84.5, 60.3, 58.7, 54.6, 48.98, 45.75, 13.7; IR (υmax) (KBr), ν/cm^−1^: 3501, 3317, 1711, 1494, 705 cm^−1^; HRMS (ESI) *m*/*z* [M + H]+ 520.1189 calcd for C28H24Cl2N3O3^+^: found: 520.1177.

Methyl (2SR,3RS,4SR,6RS)-5,5-dicyano-2-(4-bromo)phenyl-2-hydroxy-4,6-bis(4-methyl)phenyl-piperidine-3-carboxylate (**6d**) Yield: 1.42 g (87%); white solid; m.p. 144–146 °C; 1H-NMR (DMSO-*d*_6_, 300.13 MHz): δ = 7.56–7.7 (m, 4H, Ar), 7.58 (d, *J* = 8.1 Hz, 2H, Ar), 7.33–7.24 (m, 4H, Ar), 7.19 (d, *J* = 7.9 Hz, 2H, Ar), 6.15 (s, OH), 5.04 (s, H, CH), 4.32 (d, *J* = 12.4 Hz, 1H, CH), 3.63 (s, NH), 3.4 (d, *J* = 12.4Hz, H, CH), 3.09 (s, 3H, OCH3), 2.33 (s, 3H, CH3), 2.28 (s, 3H, CH3); 13C-NMR (DMSO-*d*_6_, 75.47 MHz): δ = 168.59, 144.36, 139.5, 138.67, 133.88, 133.22, 131.21 (s, 2C), 129.67 (s, 2C), 129.4 (s, 2C), 128.80 (s, 4C), 128.67 (s, 2C), 121.75, 114.15, 113.44, 84.20, 59.1, 54.6, 51.56, 49.4, 46.2, 21.26, 21.1; IR (υmax) (KBr), ν/cm^−1^: 3490, 3316, 2250, 1715, 512 cm^−1^; HRMS (ESI) *m*/*z* [M + H]+ (for 79Br) 544.1230 calcd for C29H27BrN3O3+: 544.1217.

### 3.6. Synthesis of ***7***

(2*SR*,3*RS*,4*SR*,6*RS*)-5,5-dicyano-2-(4-bromo)phenyl-2-hydroxy-4,6-bis(4-methyl)phenyl-piperidine-3-carboxylate **6d** (1 mmol) and acetic acid (2 mmol) were refluxed in methanol (8 mL) for 2 h. The mixture was maintained at –10 °C for 30 min for the complete precipitation of the product. The precipitate was collected by filtration and dried to obtain pure **7**.

Methyl (3RS,4SR,6RS)-5,5-dicyano-2-(4-bromo)phenyl-4,6-bis(4-methyl)phenyl-3,4,5,6-tetrahydropyridine-3-carboxylate (**7**) Yield: 0.18 g (90%); white solid; m.p. 235–237 °C; 1H-NMR (DMSO-*d*_6_, 300.13 MHz): δ = 7.86 (d, *J* = 8.6 Hz, 2H, Ar), 7.69 (d, *J* = 8.6 Hz, 2H, Ar), 7.54 (d, *J* = 8.1 Hz, 2H, Ar), 7.35–7.25 (m, 6H, Ar), 6.03 (d, *J* = 3 Hz, H, CH), 5 (dd, *J*_1_ = 11.2 Hz, *J*_2_ = 3 Hz, H, CH), 4.4 (d, *J* = 11.2 Hz, H, CH), 3.32 (s, 3H, OCH3), 2.35 (s, 6H, 2CH3); 13C-NMR (DMSO-*d*_6_, 75.47 MHz): δ = 169.97, 163.36, 139.4, 138.8, 136.74, 134.66, 131.96 (s, 2C), 131.93, 130.03 (s, 2C), 129.5 (s, 2C), 129.37 (s, 2C), 129.04 (s, 2C),128.72 (s, 2C), 125.20, 113.91, 112.69, 65.78, 53.25, 48.18, 47.16, 45.37, 21.22 (s, 2C); IR (υmax) (KBr), ν/cm^−1^: 2952, 2252, 1742, 1636, 1259, 500; HRMS (ESI) *m*/*z* [M + H]+ (for 79Br) 526.1125 calcd for C29H27BrN3O3+: 526.1118.

### 3.7. Synthesis of ***8***

4-flurobenzaldehyde **1h** (6 mmol), ethyl cyanoacetate **2b** (3 mmol), methyl 3-(4-bromophenyl)-3-oxopropanoate **3f** (3 mmol) and ammonium acetate (6 mmol) in methanol (10 mL) were stirred at rt for 3 days. The solvent was evaporated under reduced pressure. Compound **8** was purified by column chromatography (eluent hexane/ethyl acetate = 3/1).

5-ethyl 3-methyl (3*SR*,4*RS*,5*SR*,6*SR*)-6-(4-bromophenyl)-3-cyano-2,4-bis(4-fluorophenyl)-3,4,5,6-tetrahydropyridine-3,5-dicarboxylate (**8**) Yield: 0.42 Γ (24%); white solid; m.p. 183–185 °C; ^1^H-NMR (DMSO-*d*_6_, 300.13 MHz): δ = 7.78 (d, *J* = 8.6 Hz, 2H, Ar), 7.67 (d, *J* = 8.6 Hz, 2H, Ar), 7.45–7.18 (m, 8H, Ar), 5.92 (d, *J* = 2.8 Hz, H, CH), 4.85 (dd, *J*_1_ = 11.2 Hz, *J*_2_ = 2.8 Hz, H, CH), 4.2 (d, *J* = 11.2 Hz, H, CH), 3.9 (q, *J* = 7.2 Hz, 2H, CH_2_), 3.32 (s, 3H, OCH_3_), 0.88 (t, *J* = 7.1 Hz, 3H, CH_3_); ^13^C-NMR (DMSO-*d*_6_, 75.47 MHz): δ = 170.88, 165.02, 162.84, 162.67 (d, *J^1^*_C-F_ = 245.5 Hz, 2C), 162.38 (d, *J^1^*_C-F_ = 245.5 Hz, 2C), 137.35, 134.77 (d, *J^4^*_C-F_ = 2.9 Hz, 2C), 132.16 (d, *J^4^*_C-F_ = 2.9 Hz, 2C), 131.96 (s, 2C), 131.23 (d, *J^3^*_C-F_ = 8.5 Hz, 2C), 130.61 (d, *J^3^*_C-F_ = 8.5 Hz, 2C), 129.15 (s, 2C), 124.65, 117.61, 116.3 (d, *J^2^*_C-F_ = 21.5 Hz, 2C), 115.49 (d, *J^2^*_C-F_ = 21.5 Hz, 2C), 65.85, 63.12, 54.01, 53.01, 48.26, 47.13, 13.81; IR (υ_max_) (KBr), ν/cm^−1^: 2250, 1734, 1230, 1009, 517 cm^−1^; HRMS (ESI) *m*/*z* [M + H]^+^ (for ^79^Br) 581.0884 calcd for C_29_H_23_BrF_2_N_2_O_4_^+^: 581.0882.

## 4. Conclusions

We developed a *one-pot* pseudo-five-component stereoselective synthesis of substituted 1,4,5,6-tetrahydropyridine, utilizing aldehydes (both with electron-withdrawing and electron-donating substituents), malononitrile or ethylcyanoacetate, esters of 3-oxocarboxylic acids and ammonium acetate, which played a dual role, acting as a base and as a nitrogen source for six-membered nitrogen-containing rings. Five bonds were formed as a result of the multicomponent process. Our method allows to obtain 2-substituted alkyl (4*SR*,6*RS*)-4,6-diaryl-5,5-dicyano-1,4,5,6-tetrahydropyridine-3-carboxylates with two stereogenic centers and 3,5-dialkyl (4*RS*, 5*SR*,6*RS*)-5-cyano-2,4,6-triaryl-1,4,5,6-tetrahydropyridine-5,3-carboxylates with three stereocenters. We established the formation mechanism of 1,4,5,6-tetrahydropyridines. These compounds were formed in a sequence of fast and slow reactions, including Knoevenagel condensation, Michael addition, Mannich reaction, intramolecular cyclization, dehydration and isomerization. The polysubstituted (4*RS*,6*SR*)-1,4,5,6-tetrahydropyridine was found to be more stable than the isomeric intermediate (3*RS*,4*SR*,6*RS*)-3,4,5,6-tetrahydropyridine. The conditions of all intermediates selective preparations were specified.

## Data Availability

Not applicable.

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
