# Peer review of "Tetrahydropyridines’ Stereoselective Formation, How Lockdown Assisted in the Identification of the Features of Its Mechanism"

_molecules, 2022, doi:10.3390/molecules27144367_

Round 1

Reviewer 1 Report

The present manuscript describes diastereselective synthesis of tetrahydropyridine derivatives. The work is accurate and well written. The products were characterized by typical methods (NMR and HRMS). Some compounds, including intermediates, were characterized by single crystal X-ray analysis. The manuscript can be recommended for publication after correction of issues, described below.

1) Section describing the details of single crystal X-ray analysis is missing.

2) The crystallographic parameters in the Table 4 look acceptable, except of the Flack parameter for 7, which is 0.388(6). Please consider the possibility of racemic twinning.  

3) Please add the details of DFT calculations, mentioned in the lines 177-181, to the Experimental section. Also, please clarify the type of dispersion correction used and add the optimized structures (cartesian coordinates) to the Supplementary Information.

4) It is not clear if the statement “However, no intermediates have been isolated or identified” (Introduction, line 61) concerns literature data or present work.

5) From the phrase (line 109): “To support the validity of the proposed mechanism (Scheme 1), we monitored the reaction between...” it appears that the aim was to verify the proposed mechanism. However, while the observation of the hydroxylated intermediate 6 and isomer 7 is interesting, the fate of others (e.g. A and B, mentioned the Scheme 1) is not clear. Were they isolated and characterized? The further phrases “Thus, we found that 2-hydroxypiperidines 6 are formed as a result of a “fast” domino sequence...” (line 136) and “Proven mechanism formation of...” (Scheme 6 caption) are confusing because the manuscript seems to be focused on the last steps of reaction, but not initial ones.

Author Response

1) Section describing the details of single crystal X-ray analysis is missing.

X-ray crystallographic data and refinement details are including in Experimental Part, subchapter 3.3

2) The crystallographic parameters in the Table 4 look acceptable, except of the Flack parameter for 7, which is 0.388(6). Please consider the possibility of racemic twinning.  

Necessary comments are given in subchapter 3.3.

3) Please add the details of DFT calculations, mentioned in the lines 177-181, to the Experimental section. Also, please clarify the type of dispersion correction used and add the optimized structures (cartesian coordinates) to the Supplementary Information.

DFT calculations are added in the text (after Table 3), in Experimental Part, subchapter 3.2, and Supporting materials.

4) It is not clear if the statement “However, no intermediates have been isolated or identified” (Introduction, line 61) concerns literature data or present work.

We are talking about intermediate C, who had never been isolated previously. Formation of closest analogues of intermediate A from benzylidenemalononitriles and malononitrile or ethyl cyanoacetate was studied previously by Verboom et al [48].

5) From the phrase (line 109): “To support the validity of the proposed mechanism (Scheme 1), we monitored the reaction between...” it appears that the aim was to verify the proposed mechanism. However, while the observation of the hydroxylated intermediate 6 and isomer 7 is interesting, the fate of others (e.g. A and B, mentioned the Scheme 1) is not clear. Were they isolated and characterized? The further phrases “Thus, we found that 2-hydroxypiperidines 6 are formed as a result of a “fast” domino sequence...” (line 136) and “Proven mechanism formation of...” (Scheme 6 caption) are confusing because the manuscript seems to be focused on the last steps of reaction, but not initial ones.

Thus, a multicomponent reaction between aldehydes 1, cyano C-H acids 2 (malononitrile or ethyl cyanoacetate), esters of 3-oxocarboxylic acids 3 and ammonium acetate is a six-step domino process. At the first stage, the Knoevenagel condensation between aldehydes and cyano C-H acid occurs. Ammonium acetate is a catalyst for this reaction. Formation of cyanoolefins A under ammonium salts catalysis is known [58]. The second step of the process is the Michael addition of C-H acid 3 to the electron-poor styrene A to form the Michael adduct B. Formation of closest analogues of intermediate B from benzylidenemalononitriles and malononitrile or ethyl cyanoacetate was studied previously by Verboom et al [48]. The subsequent Mannich reaction of B, aldehyde 1 (second equivalent) and ammonia, which is formed from ammonium acetate, leads to intermediate C. The latter undergoes intra-molecular cyclization with the formation of a substituted 2-hydroxypiperidine 6, which was identified and characterized in this work for a first time. Similar sequence: Knoevenagel condensation – Michael addition – Mannich reaction – intramolecular cyclization describes by Latypova et al when studying the multicomponent reaction between 1,3-dicarbonyl compounds (two equiv.), formaldehyde and diamines with the formation of substituted bis-1,2,3,4-tetrahydropyridines [59]. None of the intermediates were isolated. Moreover, we tried to isolate C in the course of the work, but failed because in the reaction mass after 10-30 min from the reaction start were plenty of compounds (by TLC) almost impossible to be isolated due to rapid reaction rate. Polysubstituted 2-hydroxypiperidines 6 were isolated up to 87% even after stirring at rt for 40 min (see Table 2). Fifths step of domino process is C dehydration. We have established that the 3,4,5,6-tetrahydropyridines 7, 8 formation are occurs. Final isomerization affords 1,4,5,6-tetrahydropyridines 4, 5.

Reviewer 2 Report

Iliyasov et al. molecules-1750023

In this manuscript, the authors reported stereoselective multi-component synthesis of tetrahydropyridine derivatives. The chiralities were confirmed both via NMR and X-Ray crystal structures.

Comments:

(1)    The authors mentioned in line 60 and 61 “no intermediates have been isolated or identified.” Through the manuscript, still no intermediates have been isolated or identified expect for the final product as well as the pre dehydrated product. Therefore, the proposed mechanisms in Scheme 1 are still not validated without any key intermediates identified.

(2)    I had trouble finding novelties in this work compared to all the previous work self-sited in this manuscript. The conditions for forming the tetrahydropyridines reported here are basically the same compared to published previously, especially compared to reference 47.

In summary, this manuscript is not recommended for publication in Molecules as a full paper considering lacking of novelties.

Author Response

  • The authors mentioned in line 60 and 61 “no intermediates have been isolated or identified.” Through the manuscript, still no intermediates have been isolated or identified expect for the final product as well as the pre dehydrated product. Therefore, the proposed mechanisms in Scheme 1 are still not validated without any key intermediates identified.

In here we validated and proved the mechanism of the reaction due to the following:

- we slowed down the reaction rate to view the intermediates formation,

- we examined the substance 6 to give the spectrum of products through various reactions, and found out that to get products 4,5 it should go only through dehydration with formation of 7,8,

- we isolated and identified the substances 6,7,8,

- the intermediates A, B, C in the same or closest reaction types are already being studied. This can be found in the articles [58] for A, [48] for B and [59] for C. Thank you for this comment, we included this into our discussion and references, see the version updated. Moreover, we tried to isolate C in the course of the work, but failed because in the reaction mass after 10-30 min from the reaction start were plenty of compounds (by TLC) almost impossible to be isolated due to rapid reaction rate. Polysubstituted 2-hydroxypiperidines 6 were isolated up to 87% even after stirring at rt for 40 min (see Table 2).

(2)    I had trouble finding novelties in this work compared to all the previous work self-sited in this manuscript. The conditions for forming the tetrahydropyridines reported here are basically the same compared to published previously, especially compared to reference 47.

In summary, this manuscript is not recommended for publication in Molecules as a full paper considering lacking of novelties.

The crucial novelty that we introduced in the manuscript is the following:

  1. Tetrahydropyridine synthesis was performed directly with reacting aromatic aldehydes, malononitrile or ethyl cyanoacetate, esters of 3-oxocarboxylic acids and ammonium acetate. The domino process lead to formation of three С-С and two С-N bonds.

On the contrary, in the work 47 we performed multicomponent synthesis of 1,4,5,6-tetrahydropyridines from olefins, aromatic aldehydes, esters of 3-oxocarboxylic acids and ammonium acetate. The synthesis resulted in formation of two С-С bonds and two С-N bonds. Thus, we got six-membered nitrogen-containing heterocycle with two stereo centers. We proposed the mechanism, but intermediates were not found.

  1. We obtained 1,4,5,6-tetrahydropyridines both with two stereo centers (compounds 4, with use of malononitrile), and with three stereo centers (compounds 5, in reaction with ethyl cyanoacetate).
  2. Though we know existing works describing substituted piperidines multicomponent synthesis, as well as piperidine-2-ones or tetrahydropyridines from aldehydes or olefins, C-H acids and ammonium acetate or amines. However, the authors of those works only assumed the mechanisms for the multicomponent processes, passing through domino process. On the other hand, in our work we for the first time isolated the reaction intermediate product (2-hydroxypiperidine 6), and studied its chemistry. We determined that alcohol 6 dehydration proceeded with formation of isomeric substituted 3,4,5,6-tetrahydropyridine 7 or 8. And its following isomerization lead to formation of 1,4,5,6-tetrahydropyridine 4 or 5. All of the intermediates 6-8 were obtained and characterized.

Thus, a multicomponent reaction between aldehydes 1, cyano C-H acids 2 (malononitrile or ethyl cyanoacetate), esters of 3-oxocarboxylic acids 3 and ammonium acetate is a six-step domino process. At the first stage, the Knoevenagel condensation between aldehydes and cyano C-H acid occurs. Ammonium acetate is a catalyst for this reaction. Formation of cyanoolefins A under ammonium salts catalysis is known [58]. The second step of the process is the Michael addition of C-H acid 3 to the electron-poor styrene A to form the Michael adduct B. Formation of closest analogues of intermediate B from benzylidenemalononitriles and malononitrile or ethyl cyanoacetate was studied previously by Verboom et al [48]. The subsequent Mannich reaction of B, aldehyde 1 (second equivalent) and ammonia, which is formed from ammonium acetate, leads to intermediate C. The latter undergoes intra-molecular cyclization with the formation of a substituted 2-hydroxypiperidine 6, which was identified and characterized in this work for a first time. Similar sequence: Knoevenagel condensation – Michael addition – Mannich reaction – intramolecular cyclization describes by Latypova et al when studying the multicomponent reaction between 1,3-dicarbonyl compounds (two equiv.), formaldehyde and diamines with the formation of substituted bis-1,2,3,4-tetrahydropyridines [59]. None of the intermediates were isolated. Moreover, we tried to isolate C in the course of the work, but failed because in the reaction mass after 10-30 min from the reaction start were plenty of compounds (by TLC) almost impossible to be isolated due to rapid reaction rate. Polysubstituted 2-hydroxypiperidines 6 were isolated up to 87% even after stirring at rt for 40 min (see Table 2). Fifths step of domino process is C dehydration. We have established that the 3,4,5,6-tetrahydropyridines 7, 8 formation are occurs. Final isomerization leads to 1,4,5,6-tetrahydropyridines 4, 5.
